# The Growth-Arrest-Specific (*GAS*)-5 Long Non-Coding RNA: A Fascinating lncRNA Widely Expressed in Cancers

**DOI:** 10.3390/ncrna5030046

**Published:** 2019-09-17

**Authors:** Anton Scott Goustin, Pattaraporn Thepsuwan, Mary Ann Kosir, Leonard Lipovich

**Affiliations:** 1Center for Molecular Medicine and Genetics, Wayne State University School of Medicine, Detroit, MI 48201, USA; 2Karmanos Cancer Institute, Detroit, MI 48201, USA

**Keywords:** GAS5, evolution, lncRNA, cancer, ribosome profiling, cellular quiescence, small open reading frames, smORFs, riborepressor, RNA mimic

## Abstract

Long non-coding RNA (lncRNA) genes encode non-messenger RNAs that lack open reading frames (ORFs) longer than 300 nucleotides, lack evolutionary conservation in their shorter ORFs, and do not belong to any classical non-coding RNA category. LncRNA genes equal, or exceed in number, protein-coding genes in mammalian genomes. Most mammalian genomes harbor ~20,000 protein-coding genes that give rise to conventional messenger RNA (mRNA) transcripts. These coding genes exhibit sweeping evolutionary conservation in their ORFs. LncRNAs function via different mechanisms, including but not limited to: (1) serving as “enhancer” RNAs regulating nearby coding genes in *cis*; (2) functioning as scaffolds to create ribonucleoprotein (RNP) complexes; (3) serving as sponges for microRNAs; (4) acting as ribo-mimics of consensus transcription factor binding sites in genomic DNA; (5) hybridizing to other nucleic acids (mRNAs and genomic DNA); and, rarely, (6) as templates encoding small open reading frames (smORFs) that may encode short proteins. Any given lncRNA may have more than one of these functions. This review focuses on one fascinating case—the growth-arrest-specific (*GAS*)-5 gene, encoding a complicated repertoire of alternatively-spliced lncRNA isoforms. *GAS5* is also a host gene of numerous small nucleolar (sno) RNAs, which are processed from its introns. Publications about this lncRNA date back over three decades, covering its role in cell proliferation, cell differentiation, and cancer. The *GAS5* story has drawn in contributions from prominent molecular geneticists who attempted to define its tumor suppressor function in mechanistic terms. The evidence suggests that rodent *Gas5* and human *GAS5* functions may be different, despite the conserved multi-exonic architecture featuring intronic snoRNAs, and positional conservation on syntenic chromosomal regions indicating that the rodent *Gas5* gene is the true ortholog of the *GAS5* gene in man and other apes. There is no single answer to the molecular mechanism of *GAS5* action. Our goal here is to summarize competing, not mutually exclusive, mechanistic explanations of *GAS5* function that have compelling experimental support.

## 1. Early History of the Growth-Arrest-Specific (*GAS*)-5 Transcriptional Unit: Discovery in a Screen for G_0_ Genes

Proliferating cells go through periods characterized by biochemically and cytologically distinct phases known as the cell cycle (G_1_, S, G_2_, and M). Untransformed mammalian cells can be driven out of the cell cycle into a deep quiescent state by various methods, including serum starvation, which drives normal cells into a quiescent state, named G_0_ (G-zero) in the 1970s by Renato Baserga [1,2,3]. Baserga worked with a diploid human fibroblast cell line, WI-38 [4], which has a very stable karyotype. The G_0_ phase is a so-called “pre-replicative phase” distinct from the other phases in the time it takes to revive the cell from “sleep”, bringing it back into the cycle in a G_0_-to-G_1_ transition. In a search for genes whose expression is representative of this deep prereplicative state, the group of Lennart Philipson at EMBO in the 1980s sought to define the genes expressed in G_0_ cells using a subtraction library approach to clone complementary DNAs (cDNAs) from serum-starved NIH/3T3 cells—publishing a six member group of “growth-arrest-specific” or *GAS* genes (1 through 6), including *Gas5* [5]. Five of the six cDNAs encoded multiply-spliced mRNA-type coding genes, such as *Gas2*, with an open reading frame (ORF) translating into a 314-residue cytoskeletal protein highly conserved between rodents and primates, encoding a protein in the calponin-homology family [6]. The other *GAS* cDNAs (1, 3, 4, and 6) were also mRNAs like *Gas2*—all except *Gas5*.

## 2. The *GAS5* Locus Encodes a Long Non-Coding RNA (lncRNA) Gene That Contains Small Nucleolar (sno) RNAs in Its Introns While Demonstrating snoRNA-Independent Functions

*GAS5* turned out to be quite different from the other five original *GAS* genes that are typical protein-coding genes. The *GAS5* gene is not a typical protein-coding gene; it is not transcribed into an mRNA, but it gives rise to an lncRNA. Prior to the advent of genome sequencing, gene conservation in evolution was widely assumed to be the prevalent paradigm. However, in the wake of the completion of the human and mouse genome projects, and after unbiased whole-transcriptome empirical mapping efforts that generated the first mammalian gene catalogs [7], two surprising findings emerged: first, the number of non-protein-coding genes (which subsequently came to be known as lncRNA genes) exceeded the number of protein-coding genes; and second, there was a global lack of evolutionary conservation between closely related mammalian species in lncRNA gene exons, in contrast to the conservation of protein-coding genes within and far beyond mammals. In contrast to protein-coding genes, most lncRNA genes are poorly-conserved. Primate lncRNAs are rapidly-evolving and evolutionarily young [8,9], which makes them excellent candidates for molecular causation of species- and evolutionary lineage-specific phenotypes. LncRNAs, computationally defined as non-messenger RNAs that do not belong to any classical (i.e., tRNA, rRNA, etc.) non-coding RNA classes and that lack evolutionarily conserved ORFs and otherwise lack any ORFs longer than 300 nt [10] are the most abundant class of mammalian non-coding RNA genes, and their annotation in the human genome remains incomplete [11]. In contrast to small RNAs, lncRNAs are mechanistically heterogeneous, with a bewildering diversity of roles and mechanisms [12,13,14,15].

The mouse and human *GAS5* transcription units (~4 kb) are extremely complex because of the large number of exons, alternative promoter usage, and rampant alternative splicing in a small genomic space. The RNA precursor is processed to generate ten small nucleolar (sno) RNAs in the C/D-box class (*SNORD*s) encoded in introns and 13 exons that can be shuffled by alternative splicing in humans to result in 24 different mature RNA isoforms (figure 1; see also [16]). The intronic snoRNAs are just one of the four functional elements of the *GAS5* locus that makes the locus fascinating as an lncRNA. The other three are: (a) multiple exons regulated by alternative splicing that provide miRNA binding sites and that are specified in part by alternative promoter use (see blue boxes in Figure 1); (b) the riborepressor encoded chiefly in the most 3′-exon (see Section 9); and (c) the small open reading frames (smORFs) conserved between many primates (see red brackets in Figure 1). Each of these elements, which we will now discuss in detail, potentially contributes to the function of this locus and, as we will show, may do so in different ways in humans and mice.

## 3. The Evidence Supporting Orthology of Rodent and Primate GAS5 Genes

In humans, the *GAS5* gene is located on chromosome 1q25 between two coding genes—*DARS2* and *ZBTB37* situated 9.5 kb apart, which face in the same direction, but in opposite direction to *GAS5*. In mice, *Zbtb37* and *Dars2* also face in the same direction, but only 5.8 kb apart; nevertheless, *Gas5* is between the two coding genes, and like in humans, the mouse Gas5 runs in the opposite direction as the two coding genes. Synteny of the *DARS2–GAS5–ZBTB37* region between human and mice is strong evidence for true orthology between human *GAS5* and mouse *Gas5*, in line with the pillar of biology that “form follows function” (see the grey box), despite the low sequence conservation between the human and mouse *GAS5* exons [17]. Poor sequence conservation in lncRNAs does not preclude function, for at least two reasons. Firstly, the secondary structure that is functional may be conserved in the absence of detectable sequence conservation [18]. Secondly, for some lncRNAs, their presence (i.e., their transcription at a certain genomic location) may be more important than their sequence [19], especially if their function is epigenetic. Like human *GAS5*, the mouse *Gas5* gene contains snoRNAs of the C/D-box type (including *SNORD47*, *SNORD78* and *SNORD80*) interspersed in its introns.

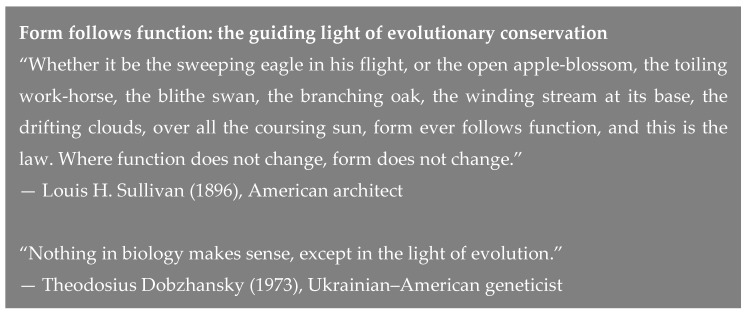


SnoRNAs predominantly guide specific types of RNA editing: rRNA modifications and spliceosomal (U) snRNA modifications that are necessary during the maturation of those RNAs. There are two major categories of snoRNAs: H/ACA-box, which mainly guide the pseudouridylation of ribosomal RNAs and small nuclear (including U1–U6, U4atac, U6atac, U12) RNAs, and C/D-box, which mainly direct 2′O-methylation of rRNA in the nucleolus, and can play a role in pre-mRNA processing [20]. In addition, snoRNAs of the C/D (*SNORD*) class have well-characterized roles in human disease [21]. The mouse and human GAS5 genes each contain 12–13 introns and encode the same nine snoRNAs (with the same number and alignable sequence) in their introns. This clearly indicates one mechanism of interspecies-conserved function as a host of these snoRNAs however, human *GAS5* encodes an additional snoRNA, designated U77, within intron 4 (Figure 1). An additional feature of the GAS5 gene is an intronic oligopyrimidine tract that argues for inclusion of *GAS5* in the 5′TOP (5′-terminal oligopyrimidine (5′TOP)) class of genes, providing an explanation for its growth-arrest role because during arrested cell growth it accumulates instead of being degraded [17].

In general, small RNAs such as microRNAs and snoRNAs are often hosted within longer protein-coding and non-coding transcriptional units, because their biogenesis precludes autonomous transcription. Hosting snoRNAs in the introns is, expectedly, not unique to the *GAS5* tumor suppressor gene, but is a feature shared with other human lncRNA genes—some of which are oncogenes: *SNHG16* (Small Nucleolar RNA Host Gene 16; also known as *NCRAN*, non-coding RNA expressed in aggressive neuroblastoma [22]), *SNHG1* [23] and *SNHG5* [24], shown in Table 1. This indicates potentially opposite functions of snoRNA repertoires encoded in the introns of oncogenic and tumor-suppressor lncRNAs that are similar in their genomic structures and conservation.

The roles of snoRNAs in the control of cell proliferation are, not surprisingly in the light of the above, gene-specific and can be both oncogenic and tumor-suppressive. Two snoRNAs (*SNORD50A* and *SNORD50B* at 6q14.3 hg19 position chr6:86,387,011-86,387,376; inside introns of the *SNHG5* gene) have been reported to directly act as breast cancer tumor suppressors [25]. Other snoRNAs such as *SNORD114-1* promote cell proliferation in leukemia [26,27]. Thus, different snoRNAs can serve as both positive and negative regulators of cell proliferation. In Table 1, the three SNHG genes at the bottom of the table (*SNHG1*, *SNHG15*, *SNHG16*) all seem to act like oncogenes in contrast to *GAS5*, which is a tumor suppressor. Perhaps the most peculiar member of the oncogenic SNHG genes is *SNHG14* [28]. The *SNHG14* gene is in the Prader–Willi region of 15q11.2, with a very long primary transcript (596 kb). This chromosome region is well known to human geneticists for its gametic imprinting [29], a phenomenon whereby maternal and paternal version of the gene are transcriptionally-active or silenced, depending on the gametic (sperm or egg) source of the autosome. Like other SNHG genes, the exons of *SNHG14* are littered with snoRNA segments, in particular of the C/D snoRNA family (*SNORDs*); including 29 copies of *SNORD116* and 48 copies of *SNORD115*. Less is known about *SNHG1*, *SNHG5* and *SNHG16* than about *GAS5*; searching PubMed for “GAS5” brings up >370 publications, and this list is growing. We focus on *GAS5* as a case study with broad implications for understanding the function of other human *SNORD* host genes that may additionally have snoRNA-independent roles.

## 4. *GAS5* Expression in Cancers

The *GAS5* transcription unit has been studied in cancer and non-cancer settings, such as hormonal signaling in normal tissues. Here we focus on its role in proliferation and apoptosis in cancer. *GAS5* expression in human tumor cells is relatively low [16,30,31] compared to normal tissues, consistent with the idea that *GAS5* exerts an anti-proliferative, tumor suppressing phenotype. In line with this tumor suppressor phenotype, plasmid-based overexpression of human *GAS5* in the estrogen-dependent breast cancer cell line MCF7 promotes apoptosis and reduces proliferation [32]. Again consistent with its tumor-suppressor role, *GAS5* transcription was found to be higher in adjacent ‘normal’ tissue than in the tumor cells probed [32].

Recent studies examined *GAS5* expression in a variety of cancers [33] including breast, bladder, ovarian, and cervical carcinomas [30,31,32,34,35,36,37,38,39,40], with a focus on cell proliferation and apoptosis. Other publications on the lncRNA *GAS5* address not only proliferation but cell migration and tumor cell invasion. Not surprisingly (given *GAS5*′s tumor suppressor role), low expression of the lncRNA indicates a poor prognosis in glioma and in breast cancer [34,41]. Likewise, *GAS5* positively regulates the transcription of miR-137 (an miRNA that suppresses triple-negative breast cancers [42]), and by doing so inhibits not only cell proliferation, but migration and invasion in melanoma cells [43]. *GAS5* has been highlighted as an inhibitor of angiogenesis as well [44]. The consensus of these reports is that *GAS5* expression is a tumor suppressor, not unlike wild-type *P53* which slows down cell proliferation and helps to promote apoptotic cell death.

## 5. The Kinase mTOR, Rapamycin, and GAS5 Transcription

The connection between the protein kinase mTOR (mammalian target of rapamycin) and *GAS5* has been known for over twenty years [17]. The kinase mTOR is a large serine/threonine protein kinase—2549 amino acids long in humans. It is the kinase in common between two multi-subunit complexes called mTORC1 and mTORC2 that enable mammalians cells to sense nutrients in the cellular milieu and to gauge the level of cellular energy [45]. When serum starvation or inhibitors of protein synthesis (such as the mTOR kinase inhibitor rapamycin) cause growth arrest, the spliced *GAS5* RNA accumulates due to its 5′-terminal oligopyrimidine tract that improves GAS5 RNA stability under these conditions [17]. In a study of prostate cancer cell lines, mTOR inhibitors enhanced GAS5 transcript levels in androgen-sensitive but not in androgen-independent cell lines, which exhibit especially low levels of endogenous *GAS5* lncRNA. This result suggests that enhancement of *GAS5* expression might have potential as a therapeutic approach in androgen-dependent prostate cancers. *GAS5* silencing induces resistance to, and ectopic *GAS5* expression confers sensitivity to, the cytostatic effects of mTOR inhibitors, thereby demonstrating a role for *GAS5* lncRNA in mTOR inhibitor action in prostate cancer [46]. In a study of mantle cell lymphoma (a B-cell lymphoma), the stimulation of expression of the tumor suppressor *GAS5* facilitated many of the cytotoxic and cytostatic effects of rapamycin and other rapalogues such as temsirolimus and everolimus [47]. These results indicate that expression of this lncRNA may delineate subgroups or molecular subtypes of cancer.

## 6. Clinical Aspects of *GAS5* Function in Breast Cancer

Cancers are heterogeneous, which has significant implications for bench research and treatment [48]. Primary and metastatic cancers differ. The stroma and inflammatory milieu further impact behavior of heterogeneous cancers. When a tumor suppressor, such as *GAS5*, is reported as important in many cancers, the implication is that there are common mechanisms to anchor both research and treatment. There are now FDA-approved therapeutic agents used to treat multiple cancers, targeting the shared molecular mechanisms of those cancers. Hence, given the broad context of the *GAS5* anti-cancer properties, future therapies that augment or rescue expression of this lncRNA might have broad relevance not just in breast cancer, but in multiple cancers.

Invasive breast cancer treatment takes first and foremost into consideration the molecular classification of the tumor. The fundamental classification of breast cancers is based on the status of intracellular receptors for estrogen, progesterone, and the cell surface receptor Her2neu—along with regional and distant metastasis. Such classification is updated at least yearly in practice guidelines (NCCN, V. 1.2019). Breast cancer treatment is guided by biopsy detection of estrogen receptor-α, progesterone receptor (PR) and Her2Neu. In addition to ER-positive, PR-positive and Her2Neu-positive classifications, there is the triple-negative class; each class is treated differently by the oncologist. Selective estrogen receptor modulators (SERMs) such as tamoxifen and raloxifene are appropriate for ER-positive tumors. Estrogen receptor positive breast cancer can recur through resistance to endocrine therapy, often due to the activation of the mTOR pathway. This specifically reduces *GAS5* levels, permitting proliferation. *GAS5* induction is thus a potential therapy.

Comparison of breast cancer tissue and surrounding noncancerous tissue confirms that *GAS5* levels are reduced in breast cancer tissue [32,49]. As expected, mTOR inhibition raises *GAS5* levels, although Pickard and Williams (2014) found that both PI3K and mTOR inhibition are needed to elevate *GAS5* levels in different cancer cell types in vitro. The clinical–translational relevance of this mechanism was supported by Phase III studies (BOLERO-2 study) using the mTOR inhibitor everolimus in hormone receptor positive advanced breast cancer patients that had previously been treated with aromatase inhibitor. The results over 18 months of follow-up showed a progression free survival (PFS) of 11.0 months vs. 4.1 months for those randomized to use exemestane with everolimus vs. exemestane with placebo [50,51]. Based on the results of the BOLERO-2 study, the NCCN guidelines now include the use of everolimus with exemestane as an option for patients who would meet the study inclusion criteria.

Patients with triple negative breast cancer (estrogen, progesterone and Her2neu receptors negative) can become candidates for endocrine therapy (anti-estrogen) given the false negative hormonal receptor status as well as differences in expression of the receptors at metastatic sites, despite the absence of receptors at the primary site. The NCCN recommends this consideration for those with localized bone or soft tissue metastasis, or asymptomatic visceral disease. This might increase *GAS5* levels.

*GAS5* is a marker for breast cancers expressing Her2neu [52]. The clinical use of trastuzumab to target the extracellular domain of Her2neu, while effective, may require additional agents when resistance develops. Li et al. [52] determined that there is decreased expression of *GAS5* in SKBR-3Tr cells that are resistant to trastuzumab. Upon treatment with lapatinib, *GAS5* levels increased and proliferation was blocked. Thus, lapatinib has been incorporated into NCCN guidelines for recurrent or metastatic Her2neu breast cancer. These examples summarily argue that the lncRNA *GAS5* might serve as a useful prognostic and diagnostic marker for a variety of cancers. In addition to serving as a biomarker, the *GAS5* lncRNA could evolve into a novel future drug target for rescue therapies.

## 7. Most Likely There Are Multiple Mechanisms, Distinct and Independent, through Which *GAS5* Expression Impacts Cell Phenotype

Although the *GAS5* lncRNA is entering its fourth decade post-discovery after its cloning as a cDNA from quiescent (G_0_) mouse fibroblasts [5], the exact molecular mechanisms through which *GAS5* lncRNA functions still present a challenge.

Evolutionary conservation can often guide us in posing a molecular hypothesis in the sense that “form follows function” (see box)—meaning that nucleotide and peptide sequence conservation suggests (but does not necessarily guarantee) function.

Chromosome location, cross-species synteny, and gene architecture can provide hints of gene function in addition to the functional content of the gene sequence. Some lncRNAs function within the confines of chromatin as ‘enhancer RNAs’ or eRNAs [53,54]. No claim has yet been made that either mouse *Gas5* or human *GAS5* lncRNA might function as an eRNA, so we will not dwell on this mechanism further. Instead, we will review other possible non-snoRNA mechanisms providing function to the *GAS5* lncRNA that have literature support.

## 8. *GAS5* lncRNA as a Scaffold Seeding the Assembly of a Nucleoprotein (RNP) Complex

Many lncRNAs, both nuclear and cytoplasmic, have been shown to serve as scaffolds that bind small subsets of specific proteins, creating an RNA–protein complex (ribonucleoprotein complex; RNP). Methods such as UV crosslinking can capture native protein–RNA interactions [55,56]. Subsequent to purification of the cross-linked RNP using biotinylated probes, proteins can be released through nuclease action, followed by mass spectrometry to reveal the proteins in the RNP. The most classic example of how lncRNA–protein complex discovery has revolutionized biology remains the chromatin-localized *Xist* lncRNA. It is argued that the 17 kb lncRNA RNP complex is the chromatin silencing machinery shutting off transcription of the majority of genes on the inactive X chromosome (X_i_), a process that accomplishes dosage compensation in placental mammals [57]. A common approach to characterization of lncRNA-seeded RNP particles employs UV (or formaldehyde) crosslinking of proteins in cell extracts, followed by pulldown of the complexes using a pool of DNA oligonucleotides tiled along the lncRNA, followed up by mass spectrometry, a technique known as RNA antisense purification followed by MS or RAP-MS [58]. This technique has been applied to *Xist* where 81 lncRNA-binding proteins were identified [59]. Using sucrose density gradient centrifugation, one early report compared the distribution of *Gas5* RNA in extracts from growing NIH 3T3 cells with that of either serum-starved cells or cells whose growth was arrested at confluency (see especially figure 6 in [17]). New approaches such as RaPID-MS [60] may prove to be useful in the identification of protein partners that bind to the lncRNA. Although these authors suggest that the heavy ribonucleoprotein (RNP) particles might be composed of ribosomes binding to the lncRNA (suggestive of translation, see below), this seems unlikely in light of other publications that fail to find that the smORF identified in mouse *Gas5* is in fact translated in mouse tissues or in 3T3 cells themselves [61,62]. An alternative possibility is that ribosomes bind to smORFs in certain lncRNAs without translating the ORF, providing a general mechanism for translation dampening via ribosome “sponging”. The rapid sedimentation of the RNPs observed might instead be interpreted as the more typical lncRNA scaffolding of specific proteins resulting in a rapidly-sedimenting RNP particle. A newish approach to the identification of open reading frames in both mRNAs and lncRNAs (known as ribosome profiling or Ribo-seq) may prove useful in understanding the question of ribosome interaction with *GAS5* lncRNA [63,64,65]. One publication using ribosome profiling addresses the issue of *Gas5* translation in mice, but not humans; this 2014 publication performs ribosome profiling on mouse embryonic stem cells, and finds no enrichment of ribosome protect fragments (RPFs) mapping to mouse exons that harbor the 39-codon open reading frame [66], lending further experimental support to the contention (which we develop in Section 13 below) that the small open reading frame in the mouse *Gas5* fails to be translated (see figure 1K in reference [66]).

According to one publication, the turnover of *GAS5* transcripts follows the paradigm of nonsense-mediated decay (NMD)—a pathway typical of aberrant messenger RNAs, and non-coding RNAs generated as a result of splicing errors [67]. This would indicate that *GAS5* is not translated beyond the mandatory “pioneer round” of translation [68]; if that is the case, ribosomes would not recurrently bind to the ORF and there would not be a translation product accumulated in cells.

Recent work suggests that the *GAS5* lncRNA directly binds to the cyclooxygenase (COX)-2 protein, but the evidence is not convincing [69]. Similarly, another recent publication suggests that *GAS5* lncRNA interacts directly with the enzyme glucose-6-phosphate dehydrogenase (G6PD), but the evidence again is weak [70]. Should these results become supported by independent studies in the future, they might provide a novel mechanism through which *GAS5* can impact energy metabolism.

To conclude, the limited interaction of ribosomes with *Gas5* transcripts derives from mouse 3T3 cells [17] and mouse embryonic stem cells [66] and does not unequivocally support true translation. Moreover, interaction of the *GAS5* lncRNA with other proteins forming an RNP is also weak. That said, one protein–RNA interaction in the *GAS5* system is strongly supported by experimental evidence: the interaction of a region near the 3′-end of human *GAS5* with four nuclear hormone receptors in the steroid receptor family, our next topic.

## 9. The Role of *GAS5* lncRNA as an RNA Mimic of the Glucocorticoid Receptor (“Riborepressor”)

A pioneering publication on this topic posits that *GAS5* lncRNA isoforms may fold into RNA secondary structures whose long RNA stems compete with the glucocorticoid receptor for binding to its DNA target [71]. The discovery was made using the yeast two-hybrid system, with the glucocorticoid receptor (GR) DNA binding domain (DBD) as the “bait”; the group identified *GAS5* as a non-coding RNA that interacts with GR. According to this report, the critical motifs seem to be encoded in a short stretch of 49 nucleotides located near the 3′-end of many of the human *GAS5* transcript isoforms (hg19; chr1:173,833,091-173,833,139), entirely within the 177 nt exon 12:5′-AUCCUCAGCCUCCCAGUGGUCUUUGUAGACUGCCUGAUGGAGUCUCAUG-3′

These authors suggest that the dsRNA stem shown in Figure 2 and Figure 3 can interact with the glucocorticoid receptor (GR) in a manner similar to interaction of the receptor with its DNA target in chromatin, the glucocorticoid response element or GRE [72], such that this region of *GAS5* RNA becomes a ribomimic of the GRE in DNA because of the sequence similarity between this double-stranded *GAS5* RNA region and the GRE in DNA. Sequence-specific DNA-binding proteins, such as transcription factors, can (in theory) bind double-stranded RNA with similar sequences. The widely-accepted paradigm asserts that glucocorticoid receptor binding to the GR ligand binding domain (LBD) in the cytoplasm leads to its translocation to the nucleus where its DNA binding domain (DBD) recognizes and binds to specific GR DNA binding sequences (GBSs, or glucocorticoid receptor element, GRE) [73]). The main idea here is that, if GR can bind its cognate sequence in *GAS5* lncRNA (instead of or in addition to the genomic GREs), then *GAS5* can repress GR-based pathways by reducing GR bioavailability; *GAS5* can titrate out bioavailable GR and prevent it from binding the promoters of GR-regulated genes in the genome. The structure depicted in Figure 2 has a calculated ΔG of −14.3 kcal/mol which is remarkable, consistent with the high (red) coloring in Figure 2. In this figure, one G nucleotide in the blue box is pointed out by a red arrow (guanine-549); mutation of this G to an A abolishes the ability of the GREM to bind the GR and (at the same time) functionally disrupts the phenotype of the riborepressor. This position is a G in humans and the other great apes, but (as pointed out by Hudson et al. [74]), the GREM is disrupted in Old World monkeys. Therefore, *GAS5* lncRNAs’ affinity for the glucocorticoid receptor protein may be a great-apes-specific function, without counterparts in other primates or in non-primate mammals such as rodents.

In recent follow-up to the Kino et al. publication [71]), authors of the follow-up publication use the term “GREM,” glucocorticoid receptor element mimic [74], to refer to the compact lincRNA–protein interaction domain in *GAS5* RNA that we diagram in Figure 2 and Figure 3. This publication utilizes fluorescence polarization to quantify the affinity of the GR for of *GAS5* lncRNA or segments of DNA (see their figure 1A of reference [74]), with K_D_ values in the sub-micromolar range. Interestingly, the authors assembled a panel of steroid receptors (the androgen receptor, the glucocorticoid receptor, the mineralocorticoid receptor, the progesterone receptor, and the estrogen receptor α), and asked which of these proteins could recognize the *GAS5* GREM. All receptors showed appreciable binding to the *GAS5* GREM with the exception of the estrogen receptor α [74]. The authors attribute the distinction between the 3-keto receptors (AR, GR, MR, and PR) and the estrogen receptor to a single key amino acid (glutamic acid, E) in the DBD of ER-α (HYGVWSCEGCKAFFKRSIQG), where the corresponding residue in the other 3-keto receptors is glycine (G). The broad implication here is that lncRNAs with sequence ribomimics of nuclear hormone receptor consensus binding sites, and of transcription factor binding sites (TFBSs) more generally, can serve as antagonists of those transcription factors. *GAS5* is the first-discovered and best-characterized member of this class. Future searches for TFBS-like sequence motifs in lncRNAs may be warranted to uncover additional lncRNAs capable of this type of function.

The GR recognizes and binds to the GRE in chromatin as a dimer [75]. As opposed to dimeric recognition of dsDNA, GR-DBD binds to RNA as a monomer and confers high affinity primarily through electrostatic contacts [76]. This contrast in how the GR recognizes dsDNA and the *GAS5* stem (Figure 2) needs to be explored in more detail, including using an approach like SHAPE-MaP [77] to experimentally explore the two-dimensional folding of the *GAS5* riborepressor, which currently is only backed by bioinformatics (i.e., folding algorithms, as in Figure 2). Alternative methods such as PARS [78] can empirically highlight stems and loops in lncRNA secondary structures. To our knowledge, neither SHAPE-MaP nor PARS has not yet been used to confirm the predicted structure of *GAS5* at the GR-binding region.

Glucocorticoid-responsive genes include both glucocorticoid-induced and glucocorticoid-repressed genes. In its well-established role in chromatin, the GR binds the promoters of various glucocorticoid-responsive genes via a glucocorticoid response element (GRE; Figure 4) including apoptosis-related genes such as GR-induced cellular inhibitor of apoptosis 2 (*cIAP2*) and serum- and glucocorticoid-regulated kinase 1 (*SGK1*). The proposed mechanism for the tumor suppressor function of GAS5 in this scenario involves the GREM out-competing the DNA-based GRE in the *cIAP2* gene, and thus driving the cell toward apoptosis when *GAS5* expression is high.

Does the glucocorticoid riborepressor element in the 3′-end of the human *GAS5* lncRNA in itself have a phenotype, without the other 11 exons? At least one publication [79] suggests that this element may operate alone to exert an apoptotic phenotype in breast cancer cells. It is intriguing that the riborepressor is encoded at 3′-end of most or all *GAS5* transcript isoforms, far downstream from the smORFs shown in Figure 1; this suggests that the GREM-dependent functions and the smORF-dependent functions may not be mutually exclusive.

## 10. Evolution of the *GAS5* Glucocorticoid Receptor RNA Mimic

The 147 nt exon 12 in human *GAS5* contains the 49 nt riborepressor, as it does in our closest relatives: bonobos (*Pan paniscus*), chimpanzees (*Pan troglodytes*), and gorillas. Next in terms of evolutionary distance from humans is the orangutan (*Pongo abelli*). Its *GAS5* exon 12 harbors a 186 nt insertion relative to humans and the other great apes, disrupting the region of the 49 nt riborepressor discovered in humans [74]. This insertion relative to humans is also present in the gibbon (*Nomascus leucogenys*) and other Old World monkeys, suggesting that the riborepressor may have evolved very recently, in the last common ancestor of gorilla, chimpanzee, bonobo and human after the orangutan split, or ~10 Myr ago. The glucocorticoid receptor gene itself is much older than primates, and is conserved broadly in placental mammals, suggesting that the riborepressor function of *GAS5* may be a relatively recent evolutionary “innovation” compared to its small open reading frames which are more deeply conserved in evolution (see below; Figure 5). There is unfortunately no publication testing for the existence of a *Gas5* riborepressor in mice, although there is a clear ortholog of *GAS5* in mice.

## 11. Interaction of *GAS5* Transcripts with microRNAs

MicroRNAs can repress mRNAs as well as lncRNAs though the RNA-induced silencing complex (RISC). Moreover, certain lncRNAs can serve as ceRNAs (competing endogenous RNAs), deflecting microRNAs from their mRNA targets—binding microRNAs using homology similar to that seen for the miRNA-mRNA interaction, and sponging the microRNAs instead of becoming degradation substrates. Interaction of *GAS5* lncRNA with a handful of microRNAs has been reported in a variety of cells [16,36,39,40,43,49,74,81,82,83,84], shown in Table 2. *GAS5* is both regulated by miR-21 and regulates miR-21 levels, in a negative feedback loop [85].

## 12. Other Plausible Molecular Mechanisms of *GAS5* Function

The long non-coding RNA *GAS5* has been shown to cooperate with the eukaryotic translation initiation factor 4E to repress c-Myc translation, and knockdown of *GAS5* increases c-Myc protein levels [89]. One completely novel molecular mechanism for *GAS5* function has recently been suggested from its role in type 2 diabetes mellitus (T2D). Using CHIP-RIP, Shi et al. [90] demonstrate that *GAS5* RNA binds a promoter element in the insulin receptor (*Insr*) gene to regulate its expression in adipocytes. Moreover, depletion of *GAS5* RNA in adipocytes inhibits glucose uptake and insulin signaling [90]. There has not been any independent evidence for or against this claim. If *GAS5* really has the capacity to bind genomic DNA at gene promoters, this would be a seminal addition to its portfolio of multifunctional mechanisms.

There is suggestion in recent publications that *GAS5* RNA remnants can be found in exosomes [91,92]. Exosomes are extracellular vesicles that are budded off from cells and derived in an endosomal compartment. Exosomes contain complex populations of mRNAs and small and large non-coding RNAs. Their packaging protects RNA from nucleases in extracellular fluids, permitting RNA to traffic between cells—a documented RNA-mediated flow of information from cell to cell [93]. *GAS5*-mediated signaling may hence be communicated by this mechanism to cells that did not originally express *GAS5* [94].

## 13. Small Open Reading Frames (smORFs), Whose Short-Protein Products Are Potentially Functional, Are Not Conserved between Mouse *Gas5* and Human *GAS5*

The majority (14 out of the 27 catalogued by [16] in their figure 1) of human *GAS5* transcript isoforms include four exons that harbor a 153 nucleotide smORF encoding a 50-amino acid smORFtide (small ORF-encoded putative polypeptide) that is conserved among primates (Figure 5). An example of one of these human *GAS5* isoforms with its conceptual translation is shown in Figure 6. One transcript isoform (*Homo sapiens* growth arrest specific 5 (*GAS5*), transcript variant 2, long non-coding RNA; GenBank NR_15251.1) a full 725 nucleotides in length features an even long smORF (222 nucleotides; positions 92–313) with a predicted smORFtide of 73 amino acids, where 23 residues are appended N-terminal to the 50 amino acids of the more common smORF, as follows:MGTRRRAGEPALRSPGTILSFRGMVLGADAVWLWIAPYGQLCPQGRMRIATEVLKSKPNSSHWHTGIRQKAGS

This longer human smORF is found in one of the many *GAS5* transcript isoforms (*Homo sapiens* growth arrest specific 5 (*GAS5*), transcript variant 2, long non-coding RNA, accession NR_152521; 725 nt). The additional 23 amino acids in this longer smORF (compared to the shorter 50-residue smORF (explicit in Figure 6; both smORFs are shown in red brackets in Figure 1) are genomically encoded in an upstream exon of 129 nt (in hg19, this 129 nt is located at chr1:173,837,892-173,838,020; see Figure 1 top line dubbed Exon 0); the 129 nt upstream exon is also present in all of the ape genomes, and most of the Old World monkey genomes that have been sequenced including macaques and baboons. Of note, this 73-residue peptide is predicted to have a very strong mitochondrial targeting signal (0.655 according to the TargetP 1.1 Server [95], and 0.68 according to SherLoc2) not shared with the shorter (50-residue peptide).

The smORF identified in the *Gas5* lncRNA isoforms from mouse (below) has no significant homology to either the long (73 residue) or short (50 residue) human *GAS5* smORF (see Table 3). BLASTP searches with the mouse query fail to yield either of the two human ORFs or any non-mouse ORFs. We have not determined whether this is due to being out of frame relative to the human, or to excessive divergence.

It is this 39-amino-acid mouse smORFtide shown in Table 3 that was used to raise an antibody by the Philipson group [61]. Their antibody recognized an 8 kD band on Western blotting of authentic mouse *Gas5* RNA translated in vitro (figure 3 in reference [61]), but failed to recognize protein from lanes loaded with protein from mouse NIH 3T3 cells, or mouse tissues including lung, brain, or heart [61]. From this evidence, the group concluded that these tissues and the 3T3 cells do not translate the mouse smORF into polypeptide. The authors qualify their conclusion by remarking that the microprotein expression level could be low (below the limit of detection), or confined to a phase of the cell cycle not represented in their NIH 3T3 cells, or simply “unstable” after translation.

The *Gas5* gene is disrupted by a frameshift mutation within its longest open reading frame in several inbred mouse strains [97], casting further doubt on the function of the smORF identified in mice by Raho et al. [61]. The *Gas5* homolog in rat at 13p22 harbors an exon 12 (79 nt), but alignment of rat and mouse shows only 66% nucleotide identity, insufficient to identify ORF conservation. For this reason, researchers in the field agree with the Raho et al. [61] conclusion that rodent *Gas5* most likely has no ORF resulting in a stable microprotein or smORFtide. Perhaps irrationally (given the well-known lack of conservation between human and mouse lncRNAs), publications on human *GAS5* seem content to assume that the coding potential of human *GAS5* mirrors the rodent *Gas5* situation. Most (if not all) publications that address the issue concur with the statement that “human *GAS5* is unlikely to encode a functionally-important polypeptide” [16]. However, because that conclusion is predicated upon rodent *Gas5*, the question of whether primate-specific *GAS5* ORFs are functional warrants further experimental investigation.

## 14. Evolution of the *GAS5* Gene in Primates

Unlike rodents, all of the great apes [*Homo sapiens*, *Pan paniscus* (bonobo or pygmy chimpanzee), *Pan troglodytes* (common chimpanzee), gorilla and orangutan] harbor clearly conserved *GAS5* ORFs, as evident from the Clustal Omega alignment of the smaller 50-codon ORF (Figure 5). In the 25 Myr separating Old World monkeys from apes [98], the *GAS5* gene ORF has diverged much less than in the 10 Myr separating mouse and rat [99], suggesting a function in African primates that maintains gene structure and the smORF in particular. It should be remarked that primates have a much longer generation time than rodents, so that one million years in rodents is many more generations than the same time span in primates [9].

The high conservation of amino acid sequence in Old World monkeys and apes is suggestive of function, but does not prove function. When subjected to a protein-folding prediction algorithm (I-TASSER (Iterative Threading ASSEmbly Refinement)), two strong α-helices are predicted in the primate *GAS5* ORF translation product (Figure 7), lending further support to the suggestion that this smORF-encoded microprotein may be real, and modestly stable when synthesized [100].

## 15. A Major Unexplored Question about GAS5 Function: Genes Responding Downstream to GAS5 RNA Level

Many lncRNAs provide high-level coordination of transcription, such as the *Xist* lncRNA that broadly silences nearly every gene on the inactive X chromosome in placental mammals. Could *GAS5* provide similar high-level regulation—influencing dozens of coding genes that together contribute to its tumor-suppressor phenotype—including genes controlling cell proliferation (such as cyclins and cyclin-dependent kinases) or apoptosis (such as the BH3-only member of the BCL2 family (Bim, Bid), caspases or PTE)? Disappointingly, no lab has approached the question using next generation sequencing (NGS) approaches like RNA-seq before and after perturbations of *GAS5* expression. It would be quite interesting to knock down *GAS5* expression (with small interfering RNA) or overexpress *GAS5* with plasmids—in an interesting cell type or multiple cell types—and then prepare RNA-seq libraries between knockdown and control and between overexpression and control. In fact, it is remarkable that such fundamental *GAS5* system perturbations have not yet been published in humans or mice. CRISPR/Cas9 could be used as well—not to delete the entire *GAS5* gene, but to alter its two small ORFs, for example through an indel that only subtly changes one of the exons involved, but creates a frameshift in the ORF. These experiments would be important for testing the hypothesis that the smORFs are relevant to the function and the phenotypic impact of *GAS5*.

## 16. Can Any Elements of the Human *GAS5* Gene Operate Independently to Yield a Tumor Suppressor Phenotype?

The *GAS5* gene in humans is prolifically multifunctional, because: (1) one gene can be transcribed and processed into several dozen different transcript isoforms; (2) this same gene harbors ten snoRNAs within its introns; (3) the most 3′-terminal exon (12) harbors a riborepressor; and (4) the first four or five exons can be spliced to form two small open reading frames with a high degree of peptide sequence conservation among human, non-human great apes, and many Old World monkeys. Several studies suggest that a small segment of the *GAS5* gene (the core of exon 12, encoding the riborepressor) can (independent of the rest of the gene) display a phenotype with a clear molecular mechanism. Thus, the riborepressor element seems to qualify as an autonomous phenotype driver. Less certain are the snoRNAs, with evidence for their independent function. To be explored in future studies is the one oddity of primate *GAS5* genes that piqued the curiosity of Lennart Philipson in the mouse *Gas5* gene—a possible functional small open reading frame (smORF). No publication has yet demonstrated that the 50-residue or 73-residue smORFs are actually translated into a peptide in cells. If stable, such a peptide could (like the case of 56-residue mitoregulin encoded by the long non-coding RNA *LINC00116*; [101,102]) have a cellular receptor that enables a ‘microprotein’ or ‘smORFtide’ to produce a cellular phenotype on its own, apart from other chemical features of the lncRNA. It would be quite intriguing to follow up on the *GAS5* smORFtides described in this review—to learn their subcellular localization, to demonstrate the peptides by mass spectrometry, and to ask if smORF-only plasmid expression can alter cellular phenotypes in a predicted manner. If either of the two proposed smORFtides can alone create a tumor suppressor function independent of the GR riborepressor, then finding their interaction partners and mechanism of action may open up a new target for the pharmacology of cancer.

## 17. Conclusions

Although the rodent *Gas5* gene is the ortholog of the human *GAS5* gene, the two genes are otherwise quite dissimilar structurally, and may have functional features that differ. One clear structural difference between rodent *Gas5* and primate *GAS5* lncRNAs concerns small open reading frames (smORFs). The 39-codon smORF in the mouse *Gas5* lncRNA investigated by the Philipson group (encoding a rather acidic protein; see Table 3) is almost certainly not translated in any cell type. Moreover, the 39-residue mouse smORF is not conserved among other species of *Mus* and is not conserved between mice and rats. On the other hand, both of the two human smORFs shown in Table 3 are not only totally different in sequence from mouse, they both have a very high predicted isoelectric point (pI of 10.02 or pI 11.55).

The human smORF50 is evolutionary conserved in at least 12 primates, with a high degree of conservation (see Figure 5). The shorter smORF is encoded over four short exons near the 5′-end of numerous human *GAS5* isoforms, including the transcript isoform NR_002578 translated in Figure 6. An unusual human *GAS5* isoform (GenBank Accession NR_152521; 725 nt) harbors a larger smORF (73 residues) split over six short exons (again near the 5′-end of the transcript; see Figure 1), with an even more basic pI (11.55).

Lastly, although *Gas5* was discovered in mouse cells as a G_0_ gene, the evidence for the tumor suppressor role of mouse *Gas5* (in *Gas5* gain-of-function or loss-of-function experiments) is disappointing in scope [82,103]. The small ORFs conserved between primates are not conserved in mice; the mouse smORF is further not conserved in rats. If there is a function to the two primate smORFs, it has not been yet demonstrated, but it is quite unlikely that the 39-codon mouse ORF has a function. One publication on the 3T3/L1 adipogenesis system suggests that *Gas5* inhibits adipogenesis [104], but careful reading of the publication pins the effect mainly to the control of proliferation of pre-adipocytes—once again the original anti-proliferative phenotype first described in 1988. Since the riborepressor demonstrated in humans [71,74] is not even conserved in Old World monkeys, it is quite unlikely that there is a functional riborepressor element in rodent Gas5 genes. Without functional smORFs and without a functional riborepressor, it is difficult to explain how the rodent *Gas5* gene can function as a tumor suppressor, as the original 1988 publication contends, unless it is the snoRNA elements that provide the tumor suppressor function of the mouse *Gas5* gene. Next-generation-sequencing (NGS) approaches like RNA-seq after *GAS5* loss-of-function and gain-of-function should definitely be undertaken (see Section 15), since they could illuminate coding genes (and pathways) responding to *GAS5* RNA abundance.

## 18. Wrap-Up

It might be something of an understatement to suggest that the rich saga of the primate *GAS5* gene still holds intrigue for cell and molecular biologists in the beginning of its fourth decade since Lennart Philipson first cloned a perplexing cDNA out of G_0_ mouse 3T3 cells using the difficult subtraction cDNA approach [5].

## Figures and Tables

**Figure 1 ncrna-05-00046-f001:**
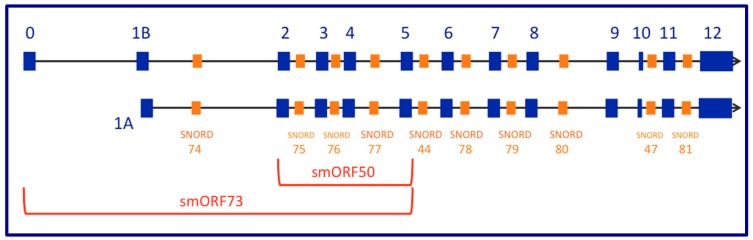
Simplified schematic of the human *GAS5* transcription unit. Shown in royal blue at the top are the 13 exons that comprise Ensemble transcript ENST00000430245.1 (or GenBank Accession NR_152521; 725 nt). The second line is the schematic of the 12 exons that comprise Ensemble transcript isoform (GenBank NR_152531; 684 nt). Orange boxes represent the location of the ten snoRNA elements encoded within introns. At the bottom are red brackets covering the four exons encoding smORF50 or the six exons encoding smORF73. Notice that exon 1B (top line; 29 nt) is not the same as exon 1A (bottom line; 32 nt). Only two of the more than two dozen human *GAS5* transcript isoforms are shown in this simplified schematic.

**Figure 2 ncrna-05-00046-f002:**
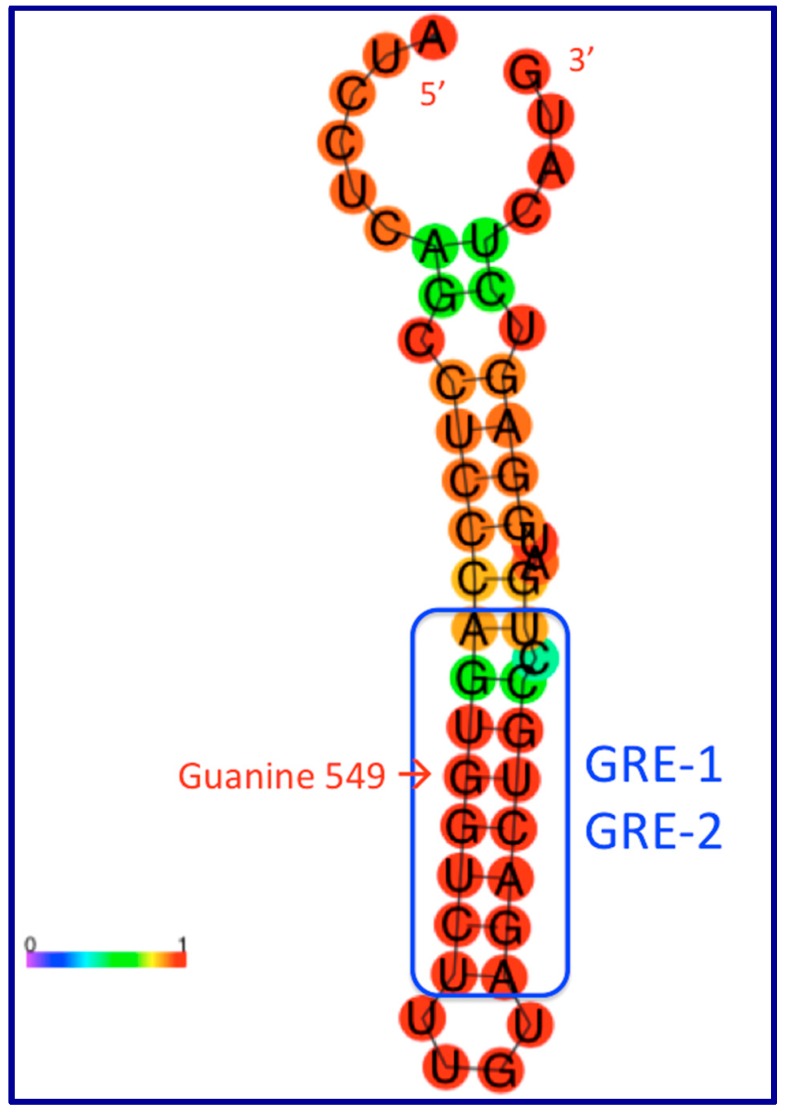
One well-supported molecular mechanism through which the human *GAS5* lncRNA acquires a phenotype involves the folding of a small (49 nt) region common to multiple RNA isoforms into a stem-loop structure with high stability--where the stems create an RNA mimic (GRE, blue box) capable of competitive binding of the glucocorticoid receptor (a known DNA-binding protein). The colored RNA structure is drawn using the web-based server found at: http://rna.tbi.univie.ac.at/cgi-bin/RNAWebSuite/RNAfold.cgi The key guanine residue (G549) described by Hudson et al. [74] is indicated with a red arrow inside the blue box.

**Figure 3 ncrna-05-00046-f003:**
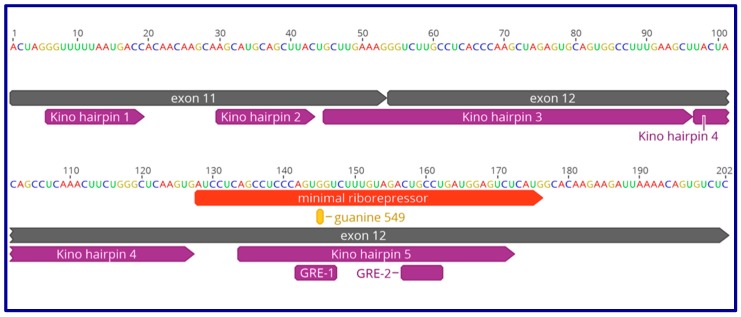
Kino et al. [71] is the first publication describing the “ribo-repressor” element formed at the 3’ end of the human *GAS5* lncRNA, present in all transcript isoforms, featuring five calculated hairpins (shown in purple). The minimal riborepressor (49 nt; shown in red above) is also depicted in a bioinformatically-calculated hairpin in Figure 2, essentially the hairpin 5 described by Kino et al. [71]. The folded RNA acts as a ribo mimic of the hormone response element in chromatin, termed “GREM” by Hudson et al. [74], who show that the riborepressor can bind the androgen, mineralocorticoid, and progesterone receptors in addition to the first-described glucocorticoid receptor.

**Figure 4 ncrna-05-00046-f004:**
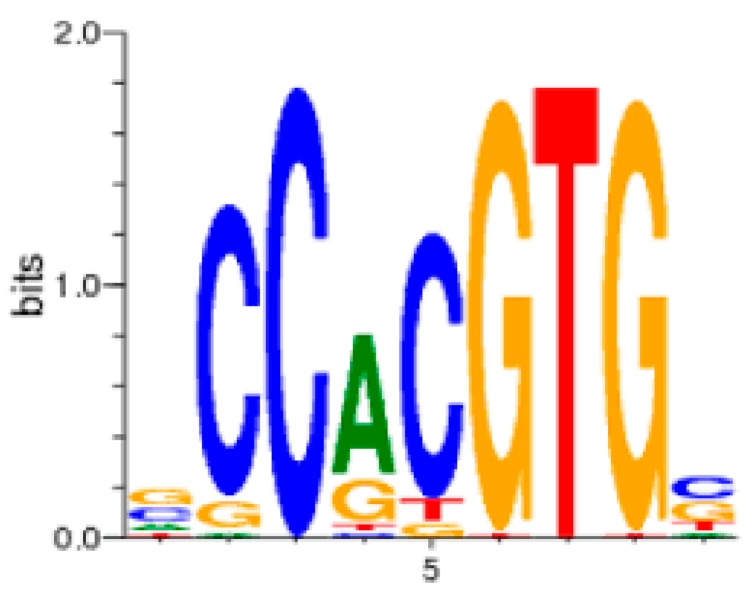
The consensus glucocorticoid response element (GRE) found in glucocorticoid-responsive genes [80]. Letter heights represent the percentage occurrence of the corresponding base at the given consensus sequence position.

**Figure 5 ncrna-05-00046-f005:**
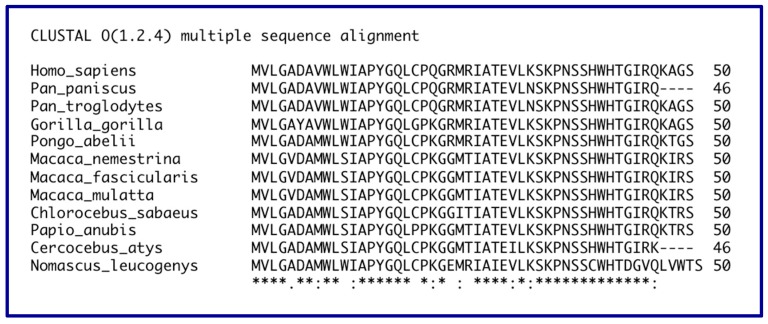
Clustal alignment of the conceptual translation of the small ORF in man and eleven non-human primates. The *GAS5* ortholog in the eleven non-human primates goes by various names including LOC103787113 (*P. paniscus*), LOC101059610 (*P. troglodytes*), LOC100936956 (*P. abelli*), LOC105491656 (*M. nemestrina*), LOC102115157 (*M. fascicularis*), LOC106994756 (*M. mulatta*), LOC103230560 (*C. sabaeus*), LOC105597571 (*C. atys*), LOC105737561 (*N. leucogenys*).

**Figure 6 ncrna-05-00046-f006:**
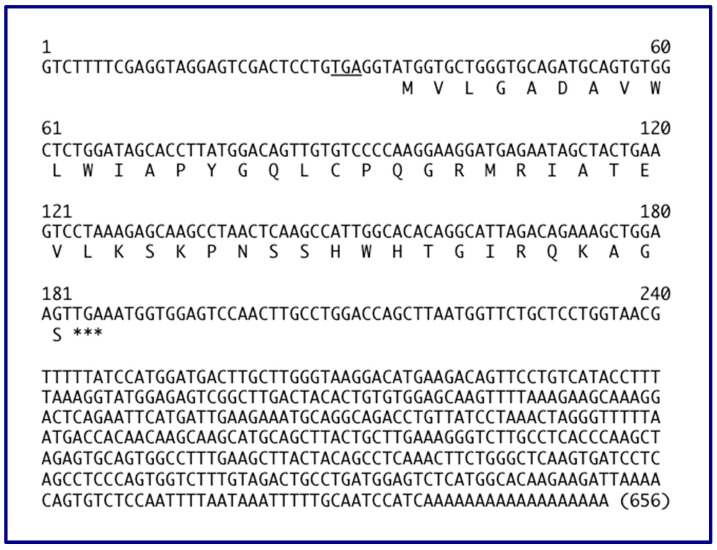
Conceptual translation of one of the 24 human *GAS5* transcript isoforms (NR_002578; 656 nucleotides) predicts a 153 nucleotide smORF encoding a 50-amino acid smORFtide. This 153 nt smORF is also encoded in 13 other human *GAS5* transcript isoforms, in *GAS5* transcripts from apes and Old World monkeys, but not from mice or rats. An upstream in-frame nonsense stop codon is underlined, a feature of protein ORFs.

**Figure 7 ncrna-05-00046-f007:**
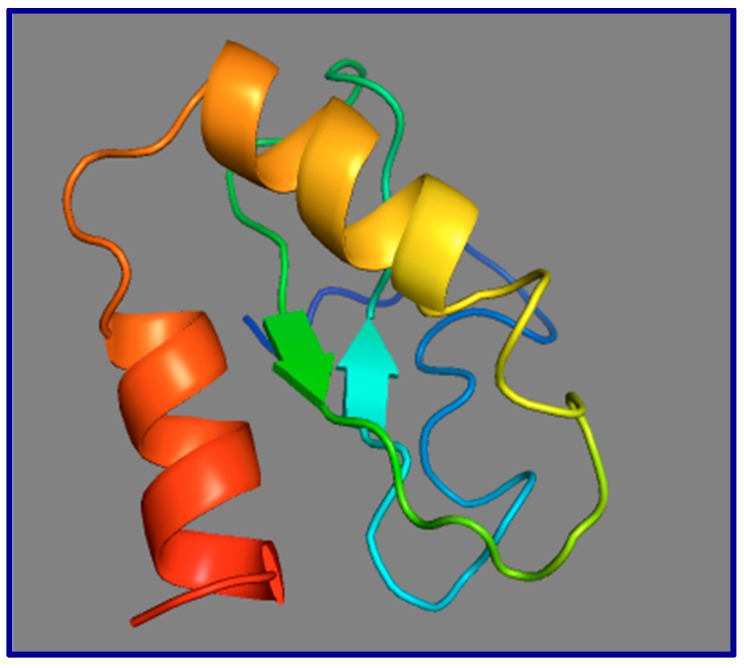
Model of the 3-dimensional structure of the human *GAS5* 73-residue smORFtide. This model was drawn as a prediction based on the conceptual translation of the human *GAS5* smORF73, using I-TASSER (Iterative Threading ASSEmbly Refinement) at the ZhangLab website at the University of Michigan. Two strong α-helices are predicted encompassing MRIATEVLKS (orange-to-yellow shading) and HWHTGIRQKA (orange-to-red shading). Beta strands are indicated as flat arrows; the N-terminal 31-amino acids are unstructured (dark blue).

**Table 1 ncrna-05-00046-t001:** Characteristics of the mouse growth-arrest-specific *(Gas)5* and human *GAS5* genes and three other human genes (*SNHG1*, *SNHG5,* and *SNHG16*) that also feature small nucleolar RNAs (snoRNAs) embedded in their introns.

Gene Name/Aliases	Phenotype	Chromosomal Locale *	Small ORFs	Effect of Estrogen	Intronic snoRNAs
**Mouse *Gas5***	Tumor suppressor	Chromosome 1 chr1:161,035,166-161,038,537	Yes	Unknown	Nine SNORD class snoRNAs
**Human *GAS5***	Tumor suppressor	Chromosome 1q25; chr1:173,833,037-173,837,129	Yes	Repression by ~60%	10 SNORD class
**Human *SNHG16* ****	Oncogene	Chromosome 17q25.1; chr17:74,553,860-74,561,430	Yes; Yu et al., [22]	Little or no effect	Three SNORD class snoRNAs
**Human *SNHG1***	Oncogene	11q12.3; chr11:62,619,460-62,623,357	Wang Q et al., [23]	Little or no effect	Nine SNORD class snoRNAs
**Human *SNHG5***	Oncogene	6q14.3; chr6:86,386,725-86,388,438	Dong et al., [25]	Very small downregulation with E2	Two SNORD class snoRNAs

* The chromosomal localization for the human genes refers to the hg19 build; for the mouse gene, we use Genome Reference Consortium Mouse Build 38. ** *SNHG16* [22] displays evolutionary conservation of its exons largely restricted to primates, like *GAS5*, even though (also like *GAS5*), it has a mouse ortholog (810032O08Rik) at the syntenic location.

**Table 2 ncrna-05-00046-t002:** MicroRNA sequences shown to interact with *GAS5*.

MicroRNA/Aliases	Site of Interaction with GAS5 lncRNA	Tumor or Cell Type	Effect Demonstrated	Reference
**miR-221/222**	Exon	Gliomas	Unknown	Zong et al., [49]
**miRNA-21**	Exon 5 *	Endothelial cells, ovarian carcinoma, cervical carcinoma	Reduced	Ma, et al. [36] Shen and She [86]; Wen et al. [39]
**miRNA-135b**	Exon	Hepatocellular carcinoma (HCC)	Sponging or translational inhibition	Yang and Jiang [87].
**miRNA-196a**	*GAS5* promoter ** or exon	Glioma, esophageal squamous cells	Downregulation of *GAS5* RNA (Wang)	Wang et al. [83]; Zhao et al. [84]
**miRNA-137**	Exon	Melanoma	*GAS5* RNA and miRNA co-regulated	Bian et al. [43]

* It is claimed [86] that the target of miRNA-21 is the 22-nucleotide region of *GAS5* (5′-ACAGGCATTAGACAGAAAGCTG-3′-OH; their Figure 1). This target lies completely within human exon 5 (the exon between *SNORD77* and *SNORD44*); this exon (our Figure 1) harbors a putative small open reading frame. In protein-coding conventional mRNAs, microRNAs usual bind 3′-UTRs, although unconventional miRNA binding to ORFs has occasionally been observed. The authors of this 2018 publication make no comment on any proposed ORF. ** It is claimed [84] that miRNA-196a (highly expressed in glioblastoma multiforme), instead of serving as a post-transcriptional regulator, can be an epigenetic regulator and binds to a region of the *GAS5* promoter which can be occupied by the transcription factor FOXO1, a TATTTT motif in the *GAS5* promoter, presumably shutting down transcription of the tumor suppressor *GAS5*. This finding is consistent with numerous reports of short RNAs serving as epigenetic regulators at promoters [88].

**Table 3 ncrna-05-00046-t003:** Conceptual translation of small open reading frames (smORFs) in rodents and primates, with molecular weights (MW) and isoelectric points (pI) for each smORF’s predicted translation product.

Species	Exons Spliced to Form cDNA	Predicted Amino Acid Sequence	RefSeq File
**Human**	Exons 2-3-4-5 *	MVLGADAVWLWIAPYGQLCPQGRMRIATEVLKSKPNSSHWHTGIRQKAGS Predicted MW 5547 daltons, predicted pI 10.02	NR_002578
**Human**	Exon 0-1-2-3-4-5 *	MGTRRRAGEPALRSPGTILSFRGMVLGADAVW LWIAPYGQLCPQGRMRIATEVLKSKPNSSHWHTGIRQKAGS * Predicted MW 8016, predicted pI 11.55	NR_152521
**Mouse**	Exon 3-4-5-6 **	MKAYEDSSGSWITERAQCARIEDQKMKWWSLRLDRQFES * Predicted MW 4767 daltons; predicted pI 6.11	NR_153812

* See Figure 1 and Figure 6. ** See mouse Dec. 2011 (GRCm38/mm10) assembly chr1:161,036,000-161,036,856. Note: MW and pI values were computed using the Compute pI/Mw tool from ExPASy [96] accessible from https://web.expasy.org/compute_pi/.

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
