# Peer review of "The Growth-Arrest-Specific (GAS)-5 Long Non-Coding RNA: A Fascinating lncRNA Widely Expressed in Cancers"

_ncrna, 2019, doi:10.3390/ncrna5030046_

Round 1

Reviewer 1 Report

In this review article, the authors tried to describe different aspects of growth-arrest-specific (GAS)-5 long noncoding RNA, describing both the biology and functions.

Although the topic is unique, nevertheless the manuscript needs to be re-structured and be written with more clarity. Authors mix the bit of everything, organization of the content is more fuzzy where events have been placed randomly. Moreover, from the organization of the manuscript it is unclear what ''a puzzling long noncoding RNA widely expressed in cancers’’ stands for?

In its current form, the manuscript is not reader friendly, for which authors need to re-organize and simplify the content.

Specific comments:

The introduction about long non-coding RNA is missing from introduction section. I suggest authors, before describing early history of GAS lncRNA, first introduce non-coding RNAs in general and the long non-coding RNAs, and finally GAS lncRNA. Authors write ''Evolution of the Gas5 gene in primates’’ as a last part of this manuscript. Perhaps it is better to move it in early chapters of the manuscript. E.g. it can be placed after ''early history’’. In the title, authors refer it as ''puzzling long noncoding RNA’’. They should provide readers a clear understanding how it stands puzzling, and what authors suggest to solve the puzzle? At several occasions the manuscript writing style appears to be bit philosophical. Indeed, it is authors choice how they want to present the manuscript; however, they may wish to make writing style more scientific. Additionally, some words give the impression that authors want to emphasize readers on certain things, by using fancy words.

Few examples.

''Fascinating’’. ''top molecular geneticists’’ – I believe writing simply molecular geneticists will be okay. Remove the word top. ''competing explanations’’. Simplify it. ''each has compelling experimental support’’. With experimental support will be okay.

And several others. Re-read and remove or balance such wordings throughout the manuscript.

Heading 6 (page 5): ''Precisely how does GAS5 lncRNA function? Exploration of molecular mechanisms’’. This heading is misleading. It gives no information. Where are the explored mechanisms? If no, then why to create such a chapter/heading? Heading 7 (page 6): authors write ‘’Most likely there are multiple mechanisms…’’. What is this most likely? Genes are borne on chromosomes. This is old fashioned description. In fact, genes are part/segments (nucleotide sequence) of chromosomes, but are not borne on chromosomes. Page 6, authors write that ''No claim has yet been made that either mouse Gas5 or human GAS5 lncRNA might function as an eRNA, so we will not dwell on this mechanism further’’. Why authors need to discuss or mention the non-existing proofs, simply delete such statements throughout the manuscript. Also, the following statement ''Instead, we will review three other possible mechanisms providing function to the GAS5 lncRNA that have literature support’’. Perhaps, authors do not need to tell readers, instead readers will decide whether the review content itself has scientific evidence. Delete such statements. While authors write [see especially Fig. 6). Please remove emphasizing on certain things. Simply cite the Fig. 6. And remove the word ‘’see especially’’. Page 11. Authors write ''This finding is so recent that there have been no publications in support or contradiction’’. What it stands for? Better to avoid such statements. I suggest to delete it.

Others.

Abstract is too long, and too fuzzy. I suggest authors to make is short and clearly but precisely tell readers what it is about. GAS5 gene in man → in human.

Author Response

RESPONSE TO REVIEWERS: ncrna-571779

Reviewer 1:

The key criticism of Reviewer 1 was taken to heart in our revision—the original submission seemed “fuzzy” at points and logically disjointed, in need of significant re-write and rearrangement of topic order. The complaint about the expression “puzzling long noncoding RNA” has been addressed by substituting the word “puzzling” In the title with the adjective “fascinating”, which truly fits the GAS5 locus with its multiple mechanisms of action (miRNA sponge, riborepressor, intronic snoRNAs and primate-conserved small open reading frames). In compliance with a request to “introduce” the lncRNAs, we do so up-front in the first sentence of the abstract, at the same time trimming 65 words from the abstract (now moved into section 2). The reviewer’s objection to the adjective “top” to describe molecular geneticists has not been implemented. Although the majority of publications on the GAS5 locus are from small, obscure groups—the fact is that Lennart Philipson’s group contributed three publications on the topic; Philipson was long-time director of EMBO (11 years); his passing in 2011 provided Nobelist David Baltimore the opportunity to review Philipson’s contributions to science in a full page obit in PloS Biology lauding Philipson as a “warrior”. The GAS5 locus has also attracted the interest of Joan Steitz, HHMI Investigator at Yale, Member of the US National Academy of Science, and Lasker Award recipient for her contributions to RNA biology and chemistry—unarguably qualifying Prof. Steitz as “top” in her field. We softened the word “top” to “prominent”.

Other suggestions of Reviewer 1 have been taken quite seriously, including deletion of superfluous language like “'This finding is so recent that there have been no publications in support or contradiction’’ (see newly-numbered section 12).

Taking to heart Reviewer 1’s unease about logical flow and order, we have especially attempted to streamline language (evidenced in the Track Changes) and improve the logical flow and presentation in the manuscript in general (above and beyond the more narrow scope recommended by Reviewer 1).

In the first paragraph of section 2, we have added a short narrative about lncRNAs in general, including addition of new publications cited.

At the end of Section 2 (manuscript page 3), we now outline very clearly the four functional elements of the GAS5 locus, which should help to bring clarity to the review in general.

In compliance with the Editor’s request, we implemented the “Track Changes” feature of MS Word to indicate edits.

Reviewer 2 Report

This is an excellent summary of what is known about an intriguing long noncoding RNA, GAS5, which was first identified as being one of a group of “growth-arrest-specific” transcripts, which is a host transcript for a number of small nucleolar RNAs, with uncertain obvious protein-coding potential.

I have a few comments and suggestions that I hope may be useful to improve the article.

It is stated in the Abstract that “LncRNAs do not function by hybridizing to nucleic acids …”. I do not think that there is any evidence for this assertion, but rather is quite possible, even likely, as a means of targeting Gas5 (and proteins with which it may scaffold / be complexed) to specific locations in DNA. I think it would be best to float this possibility, rather than deny it, or at least to remove the statement.

The abstract also states that “The evidence suggests that rodent Gas5 function and human GAS5 function may be very different, despite the multi-exon architecture featuring (the same 9) intronic snoRNAs, and positional conservation on syntenic chromosomal regions that st(r)ongly argue that the rodent Gas5 gene is the true ortholog of the GAS5 gene in man and other apes …” Again, I think that this is over-reach, and would suggest a more inclusive interpretation that its major function(s) are similar but that there may be some lineage-specific differences – as posited in the Conclusions on p. 15.

Most of the studies of Gas5 have been descriptive – mainly its expression in cancers. I might have missed it, although the statement at the end (p.15) that “It would be quite interesting to knock down GAS5 expression (with small interfering RNA) or overexpress GAS5 with plasmids …” suggests that I did not. So to pose the question: Have there been any studies which target the mature (intronless) Gas5 host RNA to try to distinguish its functions from those of its enclosed snoRNAs. If so, there should be highlighted. If not, that needs to be highlighted as well.

Author Response

Response to Reviewer 2:

Reviewer 2 objected to the statement in the Abstract that “lncRNAs do not function by hybridizing to nucleic acids”. This statement is now deleted.

We have also softened the language concerning fundamental differences between the mechanism(s) of action of the rodent and primate GAS5 genes.

In compliance with the Editor’s request, we implemented the “Track Changes” feature of MS Word to indicate edits.

Reviewer 3 Report

The topic of this review is of a great interest. Yet the authors should do careful proofreading before the submission of any paper to any journal. My first impression was that the wrong version (just the very first draft) of the manuscript was accidentally uploaded.

I recommend the authors to read more about snoRNAs: their classification, function and terminology.

When I saw snoRNAs mentioned in a context of GAS5, I expected to find a link between GAS5 function as a tumor suppressor and twoSNORDs encoded in this locus that were reported upregulated in colorectal tumors. This is what is really puzzling. Yet, snoRNA story went nowhere.

Figure 4 shows panel D of Figure 8 from Kino et al (2010) without any proper indication. There is nothing wrong if previously published figures are used in a review paper. But you cannot just copy and paste. This alone raised a major concern about this review. 

Author Response

Response to Reviewer 3:

The reviewer is correct that one of the panels in Fig. 4 was a partial copy-and-paste from a published figure, in violation of copyright. This portion of Fig. 4 is now deleted; the only part of Fig. 4 remaining was generated in our lab with the help of a licensed copy of the Geneious software package (version R11). Fig. 4 also has an expanded legend.

Reviewer 3 also suggests that “the snoRNA story went nowhere”. The snoRNA angle is addressed now in new section 3 (“The evidence supporting orthology of rodent and primate GAS5 genes”).

In compliance with the Editor’s request, we implemented the “Track Changes” feature of MS Word to indicate edits.

Reviewer 4 Report

This interesting manuscript describes the involvement of lncRNA GAS5 in various types of cancer and offers a very comprehensive review on a very specific topic for which not many reviews are available.

My major suggestion is that the review lacks a section about exosomal GAS5.

Author Response

Response to Reviewer 4:

The reviewer asks about exosomes. Three references have now been added addressing this topic, now in new Section 12 (“Other mechanisms…”).

In compliance with the Editor’s request, we implemented the “Track Changes” feature of MS Word to indicate edits.

Round 2

Reviewer 1 Report

The authors have addressed the points and have improved the manuscript. 

Only minor comment. Please remove the word "fascinating" from the title. 

Author Response

Between the original submission and the first revision, we agreed to change the word “puzzling” in the title to “fascinating.”

We are however reluctant to remove the adjective “fascinating” from the title. We are a pioneering lab in the lncRNA field with nearly two decades of experience in, and numerous publications on, lncRNA evolution and function. Within the context of that experience, we indeed find GAS5 to be one of the more “fascinating” lncRNA genes, out of the nearly ~ 20,000 human lncRNA genes catalogued to date. If it weren’t fascinating, we wouldn’t have devoted a review to it.

For these reasons, and because critiques at the v2->v3 stage that negate or oppose the responses to critiques at the v1->v2 stage are generally counterproductive, we are not making this change and we have stated these reasons to the editor.

Reviewer 3 Report

In this revised version the authors still paid little if any attention to grammar, spelling, punctuation and terminology. I believe, if the authors read carefully the text and figures, they can spot and correct all grammatical issues and misspelling. 

Some suggestions to improve the manuscript (this is not an exhaustive list of all corrections needed!):

Abstract as a whole is very lengthy. The sentence that occupies lines 16-22 is especially difficult to follow. I think numbering may help: '… (1) serving as “enhancer” RNAs regulating nearby coding genes in cis, (2) serving as scaffolds to create ribonucleoprotein (RNP) complexes, (3) serving as sponges for microRNAs, (4) acting …' . This sentence also has grammatical issues.

The authors are mixing established nomenclature of vertebrate snoRNAs (SNORD, SNORA with numbers) and their alternative names from early publications (U with numbers). This is confusing, especially when the main point was to emphasize ‘orthology’.

Lines 118-120. The sentence, as it is written now, is misleading. Do the authors know that minor spliceosomal snRNAs U12, U4atac, U6atac are also modified by snoRNAs? Furthermore, snoRNA targets are not limited to rRNAs and spliceosomal U snRNAs. Thus, the correct statement would be that snoRNAs predominantly guide rRNA and spliceosomal U snRNA modifications. While yeast U snRNAs are not 2’-O-methylated, in higher eukaryotes such as human and mouse (main focus of this review), spliceosomal snRNAs contain numerous 2’-O-methylated positions, and these modifications are positioned by box C/D snoRNAs.

The order of the last 3 sections looks mixed up: 16. Conclusions, 17. Can any elements of the “baroque” human GAS5 gene operate independently…? 18. Wrap-up. And I would not call GAS5 gene ‘baroque’.

Figure legends are embedded in most figures. At the same time, it is not obvious what is figure 3 legend. Figure 5 has 2 panels that are separated as two different figures.

Figure 1. It is not clear why figure legend says that red bracket covers five exons encoding smORF73. I can count 6 exons in the top transcript. Also, exon numbers are shifted, they appear in intronic regions containing snoRNAs. This would be very easy to spot and fix if authors did proofreading before submitting the revised version.

Figure 4. Kino (2010) should be Kino et al. (2010)

Figure 5B. I am not a color-blind person yet I find it hard to discriminate ‘royal blue’ from dark blue, ‘Robbins egg blue’ from other shades of cyan. Furthermore, beta strands are shown as arrows (no need to specify a particular shade of cyan), whereas the unstructured N-terminal tail is colored from dark blue to cyan (‘royal blue’ highlights only a small portion).

Also, there is a typo in the last sentence of Figure 5B legend.

Author Response

The remarks of Reviewer 3 about the abstract have been addressed by adopting the suggestion of numbering the mechanisms, and by fixing the grammatical problem.

Regarding the snoRNA nomenclature in Figure 1: We have replaced U#s with SNORD#s in Figure 1. The numbers are the same for each ortholog in human and mouse under both the old U system and the new SNORD system, however. The alignment of blue numbering of the exons has also been addressed.

Addressing the comment about lines 118-120: We purposely omitted the AT-AC splicing / U12 spliceosomal pathway because it is extremely rare. This is not a comprehensive review of snoRNAs. It is a review of what is known about GAS5. We simply mention them to the extent that such background is necessary to review GAS5 as a snoRNA host gene. We have made the requested changes in this paragraph.

The reviewer’s concerns about the word ‘baroque’ have been addressed by eliminating the use of the expression. Moreover, we have performed the requested changes and switched the order of sections 16 and 17.

Regarding the legends to Figures 3 and 5A/5B: Figure 5A is now 5, 5B is now 6, 6 is now 7. We have added a one-sentence legend for figure 3.

Specifically addressing concerns about Figure 1: We recounted the exons that comprise smORF73, and Reviewer 3 is correct—there are six, not five. We have made that change to the legend (six, not five). The reviewer comments about exon numbering; in response, we adjusted the blue numerals at the top of the figure to better align to the exons (blue boxes).

The reviewer’s comments about the Kino reference in Figure 4 are adopted.

The reviewer asks about the coloration and legend of Figure 5B and its legend; changes have been adopted.

Thank you for the thorough review. We have fixed the typo (“he” -> “the” in the legend of what is now Figure 6).